


# Part 2: Joint multifractal analysis of available wind power and rain intensity from an operational wind farm

Jerry Jose[a], Auguste Gires[a], Ernani Schnorenberger[b], Yelva Roustan[c], Daniel Schertzer[a], Ioulia Tchiguirinskaia[a]

[a]*HM&Co, École des Ponts ParisTech, 77455 Champs-sur-Marne, France*
[b]*Boralex, Lyon, France*
[c]*CEREA, École des Ponts, EDF R&D, Île-de-France, France*

Correspondence to: Jerry Jose (jerry.jose@enpc.fr)

## Abstract

Wind power production plays an important role in achieving UN's (United nations) Sustainable development goal (SDG) 7 - affordable and clean energy for all; and in the increasing global transition towards renewable and carbon neutral energy, understanding the uncertainties associated with wind and turbulence is extremely important. Characterization of wind is not straightforward due to its intrinsic intermittency: activity of the field becomes increasingly concentrated at smaller and smaller supports as the scale decreases. When it comes to power production by wind turbines, another complexity arises from the influence of rainfall, which only a limited number of studies have addressed so far suggesting short term as well as long-term effects. To understand this, the project RW-Turb (https://hmco.enpc.fr/portfolio-archive/rw-turb/; supported by the French National Research Agency, ANR-19-CE05-0022) employs multiple 3D sonic anemometers (manufactured by Thies), mini meteorological stations (manufactured by Thies), and disdrometers (Parsivel$^2$, manufactured by OTT) on a meteorological mast in the wind farm of Pays d'Othe (110 km south-east of Paris, France; operated by Boralex). With this simultaneously measured data, it is possible to study wind power and associated atmospheric fields under various rain conditions.

Variations of wind velocity, power available at the wind farm, power produced by wind turbines and air density are examined here during rain and dry conditions using the framework of Universal Multifractals (UM). UM is a widely used, physically based, scale invariant framework for characterizing and simulating geophysical fields over wide range of scales which accounts for the intermittency in the field. Since rated power acts like an upper threshold in statistical analysis of empirical wind power, efforts were made to use the theoretical available power as proxy to see the difference. From an event based analysis, differences in UM parameters were observed between rain and dry conditions for the fields illustrating the influence of rain. This is further explored using joint multifractal analy-



sis and an increase in correlation exponent was observed between various fields with an increase in rain rate. Here we also examine the possibility of difference in power production according to type of rain (convective or stratiform) as well as various regimes of wind velocity. While examining time steps according to wind velocity, power curves showed different regions of departure from state curve according to the rain rate.

*Keywords:*
wind, rainfall, disdrometer, multi fractal, wind power

## 1. Introduction

Wind energy is seen as the forerunner in renewable energy sector whose rapid growth (4 times greater than the current rate) is highly desired for a sustainable future (where 57%
of global power supply is renewable by 2030, from 26% in 2019) that assures climate protection (, IRENA). Wind power production also plays an important role in achieving UN's (United nations) Sustainable development goal (SDG) 7 - affordable and clean energy for all. According to the IEA 2020 wind overview, global wind power capacity has increased by 14 %, with annual installations increasing by 54 % or 60 GW (IEA, 2020). This is pro-
jected to increase as UN high-level dialogue on Energy in 2021 (UN, 2022) has called for global doubling of annual investment in renewable energy and energy efficiency by 2025 (triple by 2030 creating 60 million jobs worldwide).

Modern wind turbines extract power from wind in the atmosphere and convert it into electricity that can be stored as well as distributed to locations of use via power grids.
Popularly known term 'wind mill' refers to the historic usage where wind power was converted to mechanical energy at the location of usage (Manwell et al., 2010). According to WindEurope (EWEA previously), an average offshore wind turbine (of capacity 2.5-3 MW, Vestas V90 used in this study falls under this category) can produce more than 6 million kWh a year which is enough for 1,500 average EU households. As per their estimation, by
2050, wind power production is expected to meet 50% of EU's energy demands (EWEA, 2012). In the context of France, wind alone accounts for one third of total renewable power production in 2021 (Jørgensen and Holttinen, 2022) which is set to increase as the country targets to have 50 offshore wind farms by 2050 through simplified legislation (Engie, 2022). One of the results from Cai and Bréon (2021)'s evaluation of wind power potential
in France is that climate change will not significantly impact the statistical properties of mean load factor, thus making wind a reliable energy source in these changing times.

Small scale fluctuations and intermittence in wind makes its characterization difficult as a field, which in turn shows further spatiotemporal variability. This along with the atmospheric turbulence (more complicated owing to hub location near the boundary layer)
are transferred to the power produced. To account for this, a common practise is to use a coarser parameter such as Turbulent Intensity (standard deviation of wind speed divided by


mean wind speed over 10 min) which does not capture neither the above said complexities in smaller scales nor effect of external turbulent factors such as rain (Johnson, 2004). Only a limited number of studies have tried to address the effect of rain in power production so

far. An earlier study by Corrigan and Demiglio (1985) reported a reduction in power production (20 % to 30 %, using a 38 m diameter two-blade turbine); this was later confirmed experimentally (Al et al., 1986). Cohan and Arastoopour (2016) (improving upon Cai et al. (2013)) examined the effect of rain on wind turbine blade aerofoil using multiphase (air as volatile and rain as liquid) computational fluid dynamics (CFD) and reported high sensi-

tivity to performance in lower rain rates till rain rate is high enough to immerse most of the aerofoil surface underwater. Some positive influence of rain was also reported such as cleaning of blades (Corten and Veldkamp, 2001) increasing power production. Rain can also have long-term effects as the kinetic energy of impacting raindrops can cause leading-edge erosion (LEE) on turbine blades reducing their aerodynamic performance; this in turn

results in lower annual energy and increased downtime (Keegan et al., 2013).

It is hence of interest to quantify the effect of rainfall on wind power (theoretically available and operationally measured). The widely used scale invariant framework of Universal Multifractals (UM) is of interest to characterize wind and its correlation with other atmospheric fields (Schertzer and Lovejoy, 1987). Calif and Schmitt (2014) illustrated the

75 intermittent and multifractal nature of turbulent wind speed and aggregate power from a wind farm over a wide range of scales and showed a coupling between using generalized correlation function (GCF) based joint multifractal description (Meneveau et al., 1990). The specific framework of UM was used previously Fitton et al. (2011, 2014) for studying the scaling behaviour and multifractal properties of wind velocity and torque fluctuations.

Here, continuous high-resolution (100 Hz) measurements of 3D wind velocity along with other atmospheric fields (and rain) from a meteorological mast located at a functional wind farm (Gires et al., 2022) were subjected to multifractal analysis in a two fold analysis. The first part consisted of multifractal characterization of the fields using UM; this was followed by characterization of correlation using Joint MultiFractal analysis (JMF) which is derived

off UM (Gires et al., 2020).

Details of data collection and quality are presented in the second part of the upcoming section on data and methods; the first part of this section briefly recapitulates the framework of UM and JMF. In the first part of section 3, individual UM analyses of fields are presented along with the biases encountered. In the second part of section 3, various fields

are analyzed jointly (using JMF) and the correlations obtained between various fields are discussed along with possible biases. In section 4, the influence of rain type as well as that of wind direction on power production are discussed. Section 5 concludes the study and summarizes the results.



## 2. Methodology and data

### 2.1. Scaling analysis and UM framework

Spectral analysis is widely used for characterizing scaling properties; here, the second-order statistics of rain in the frequency domain were examined for power-law scaling as follows (Mandelbrot, 1982; Schertzer and Lovejoy, 1985).

$$E(k) \approx k^{-\beta} \tag{1}$$

where $k$ corresponds to the wave number and $\beta$ is the spectral exponent.

However, to fully characterize the complexity of the process, across its intensities and spatiotemporal variation, information on higher and lower-order statistics is required. For this, we use Universal Multifractals (UM) which relies on the assumption of the field being generated by an underlying cascade process with conserved statistical properties at each scale, while inheriting the scale invariant properties of Navier-Stokes equations (Schertzer and Lovejoy, 1987, 1989; Schertzer and Tchiguirinskaia, 2020). In this framework, the probability of a field exceeding a particular threshold across all scales is captured using the scale-invariant notion of singularity ($\gamma$) and for a multifractal field this scales according to the resolution ($\lambda = L/l$, i.e. the ratio of $L$, the outer scale, to $l$, the observational scale) with corresponding fractal codimension as the scaling exponent, $c(\gamma)$:

$$p(\varepsilon_\lambda \geq \lambda^\gamma) \approx \lambda^{-c(\gamma)} \tag{2}$$

This relation implies that statistical moments $q$ of the field scale with resolution (Schertzer and Lovejoy, 1987, 1988) with moment scaling funciton $K(q)$ as:

$$\langle \varepsilon_\lambda^{\ q} \rangle \approx \lambda^{K(q)} \tag{3}$$

$K(q)$ and $c(\gamma)$ are equivalent functions, related through Legendre transform (Parisi et al., 1985) and they fully characterize the variability of the field across all scales. For a conservative field in UM framework, $K_c(q)$ can be fully determined with the help of only two parameters with physical interpretation, multi-fractality index $\alpha$ and mean intermittency codimension $C_1$. This yields:

$$K_c(q) = \begin{cases} \dfrac{C_1}{\alpha - 1}(q^\alpha - q) & \alpha \neq 1 \\ C_1 q \ln q & \alpha = 1 \end{cases} \tag{4}$$

$C_1$ measures clustering of average intensity across scales ($C_1 \in [0,1]$ for 1 dimensional fields); when $C_1 = 0$ the field is homogeneous with little variability. $\alpha$ measures how this clustering changes with respect to intensity levels ($\alpha \in [0,2]$); higher the value of $\alpha$, higher the variability, with $\alpha = 0$ being a monofractal field where intermittency of extreme is same as that of mean.




For a non conservative field $\psi_\lambda$, i.e. a field whose average ($\langle \psi_\lambda \rangle$) changes with scales, a non-conservative parameter $H$ is used in the expression of scaling.

$$\psi_\lambda = \varepsilon_\lambda \lambda^{-H} \tag{5}$$

where $\varepsilon$ is a conservative field characterized with $C_1$ and $\alpha$. For a conservative field, $H$ = 0. For a non-conservative field with positive $H$, fractional differentiation is required to retrieve a conservative field. Similarly, from a non-conservative field with a negative value of $H$, the conservative field is retrieved through fractional integration. $H$ is related to the spectral slope $\beta$ (Eq. 1).

$$\beta = 1 + 2H - K_c(2) \tag{6}$$

The scaling behaviour of conservative multifractal fields can be examined using trace moment (TM) where log-log plot of upscaled fields against resolution $\lambda$ is taken for each moment $q$ (Eq. 3). The quality of scaling is given by the estimate $r^2$ of the linear regression; the value for $q = 1.5$ is used as reference. Double trace moment (DTM) is a more robust version of TM tailored for UM fields where the moment scaling function $K(q, \eta)$ of the field $\varepsilon_\lambda^{(\eta)}$ (field raised to power $\eta$ at maximum resolution and renormalized) is expressed as a function of multifractality index $\alpha$ (Lavallée et al., 1993).

$$\langle (\varepsilon_\lambda^{(\eta)})^q \rangle \approx \lambda^{K(q,\eta)} = \lambda^{\eta^\alpha K(q)} \tag{7}$$

From the above equation, value of $\alpha$ can be obtained as the slope of the linear part when $K(q, \eta)$ is represented for a given $q$ as a function of $\eta$ in log-log plot. Both TM and DTM techniques give reliable estimates as long as the $H < 0.5$ for the field analysed.

Since multifractal processes are generated by cascade processes, the average values can get too concentrated over a certain area leading to spurious estimates of moments above a particular value of $q$ (at $q_D$, $q$ above which $K(q) \approx +\infty$) - divergence of moments. The functions $K(q)$ and $c(\gamma)$ are also limited by the sample size of data, or rather the maximum value of scale-invariant threshold or singularity ($\gamma_s$) and corresponding moment ($q_s$). For reliable statistical estimates of the moment scaling function and hence the UM parameters, the moment orders should not be exceeded beyond $q_s$ or $q_D$.

### 2.2. Framework of joint multifractals (JMF)

Though not extensive, various methodologies were suggested and used for studying coupling (across scales) between two simultaneously measured fields from their joint moments (like moments of individual fields mentioned before, but by multiplying both fields under consideration). Meneveau et al. (1990) used joint moment exponents to examine the correlation between velocity and temperature fluctuations in the turbulent wake of a heated cylinder, and also between square of vorticity fluctuations and dissipation of turbulent velocity component. Seuront and Schmitt (2005a,b) expanded upon this by introducing



a 'generalized correlation function' (GCF, re-normalizing the joint moments) and argued the use case in effectively characterizing biological and physical coupling (using data on phytoplankton concentration, through fluorescence, and temperature at various turbulence intensities). Calif and Schmitt (2014) used GCFs to examine coupling between simultaneous data of wind speed and aggregate power output from a wind farm. Both cases used GCFs on log-normal cascades involving single parameter and linear correlation functions and explored only two specific coupling cases between fields - a proportional or a power law relation. Between two fields, the GCF is symmetrical with respect to the moment between fields; this suggests the possibility of expressing the two quantities with a simple relation of proportionality. Relying on this, Gires et al. (2020) expanded GCFs to UM providing a framework (JMF) where the related fields can be expressed as multiplicative power law combination of known UM fields. This framework not only retrieves the proportionality constants between fields but also provides an intuitive indicator that combines most of the information obtained from JMF.

Consider two simultaneously measured multifractal fields $\varepsilon_\lambda$ and $\phi_\lambda$ of resolution $\lambda$. In JMF, we can express $\varepsilon_\lambda$ in terms of $\phi_\lambda$ and an independent multifractal field $Y_\lambda$ with same $C_1$ as $\phi_\lambda$. Below, both fields are correlated with $a$ and $b$ (relative weight in combination), and $Y_\lambda$ (can be generated if we know its $\alpha$ and $C_1$). Note that $\phi_\lambda{}^a Y_\lambda{}^b$ is a single field expressed as a power law combination of $\phi$ and $Y$:

$$\varepsilon_\lambda = \frac{\phi_\lambda{}^a Y_\lambda{}^b}{\langle \phi_\lambda{}^a Y_\lambda{}^b \rangle} \tag{8}$$

Before proceeding further, it is important to state the meaning of $a$ and $b$ intuitively on correlation between fields. When $a = 1$ and $b = 0$, $\varepsilon_\lambda$ is simply equal to $\phi_\lambda$ (maximum correlation) and during the converse, $\varepsilon_\lambda$ is equal to $Y_\lambda$ with no connection to $\phi_\lambda$. Intermediate values of $a$ ($1 > a > 0$) shows progressive decorrelation between $\varepsilon_\lambda$ and $\phi_\lambda$. With $a$, $b$ and $Y_\lambda$, it is possible to characterize the correlation between two multifractal fields. Along with these parameters, JMF framework also introduces a simplified indicator of correlation, $IC_{\varepsilon\phi}$ ($\approx IC_{\phi\varepsilon}$)

$$IC_{\varepsilon\phi} = \frac{C_{1,\phi} a^{\alpha_\phi}}{C_{1,\varepsilon}} \tag{9}$$

More information on the intuitive indicator and exponents can be found in Gires et al. (2020) along with validation of the framework with real and simulated data, and a discussion on some limitations. IC is reported to be relevant for values of $\alpha$, typically greater than 0.8, which is the case for the field studied here.



### 2.3. Instrumentation, data and biases

<sup></sup>185 *2.3.1. Instrumentation and directly measured fields*

As discussed, understanding the long-term and short-term effect of rainfall on wind power production is important and the Rainfall Wind Turbine or Turbulence project (RW-Turb, `https://hmco.enpc.fr/portfolio-archive/rw-turb/`), supported by Agence Nationale de la Recherche (ANR, French National research agency in English) is designed towards addressing this with simultaneous real-time in-situ measurements of rain and wind at turbine location. To recap, RW-Turb measurement campaign (Pay d'Othe, 110 km southeast of Paris, France) consists of a meteorological mast in an operational wind farm (jointly operated by Boralex and JP Énergie Environnement) with two sets of optical disdrometers (OTT Parsivel$^2$), 3D sonic anemometers (ThiesCLIMA) and mini meteorological station at heights roughly 45 m and 80 m. The finest time-step of measurement available are 30 s, 0.01 s, and 1 s respectively. Fig. 1 briefly summarize the instrumentation and location of the meteorological mast.

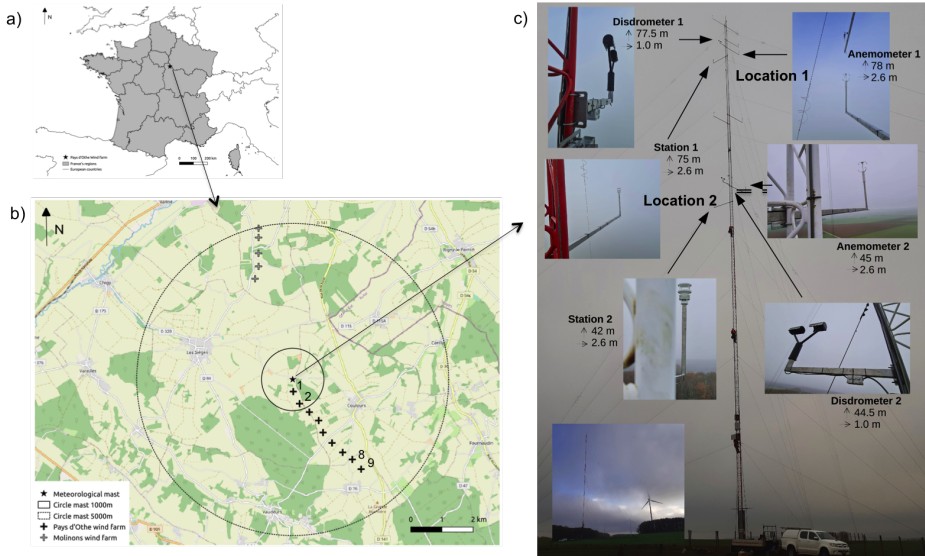

Figure 1: a) Location of the Pays d'Othe wind farm in France; b) Map of the surroundings, meteorological mast is at the centre and turbines available are numbered - 1, 2, 8 and 9; c) Summary of measurement devices on the meteorological mast and their vertical locations. Figures adapted from Gires et al. (2022).

Interested readers are directed to (Gires et al., 2022) for an overview of the campaign with data and instrumentation; a three-month-long dataset is also made publicly available


there along with the raw files and scripts required for their usage. Actual sampling rates are discussed in the next section (section 2.3.3). Daily overall information can be accessed through quicklooks at the project's web page as mentioned before, `https://hmco.enpc.fr/portfolio-archive/rw-turb/`. Quicklook for a rainy day (08/04/2022) is shown in Fig. 2. Temporal evolution of rain rate, drop size, dropsize - velocity curve, and DSD curve

highlighting influence of raindrop volume are shown in first column (in that order). Except for the first panel (Cumulative rainfall depth vs. time), the second column deals with wind velocity. Total horizontal wind ($\sqrt{u_x^2 + u_z^2}$ vs. time at one min time step) for anemometers and stations are shown in second panel of this column. The last two panels show wind rose (using the horizontal wind measurements - $u_x$ and $u_y$) and vertical wind ($u_z$ at one min

time step) from the anemometers. The missing time steps for all the devices for the day are shown in third column; the remaining panels of third column consists of temporal evolution of temperature, pressure and relative humidity from station (also temperature from anemometer). The last column consists of temporal evolution and power curves (power vs. velocity, theoretical curve -i.e. power state curve provided by manufacturer- in red) for

Turbine 1 and Turbine 9 (the closest and the farthest from the mast shown for illustration). The turbine data is not available in online quicklook or in data paper since it is private information owned by Boralex.





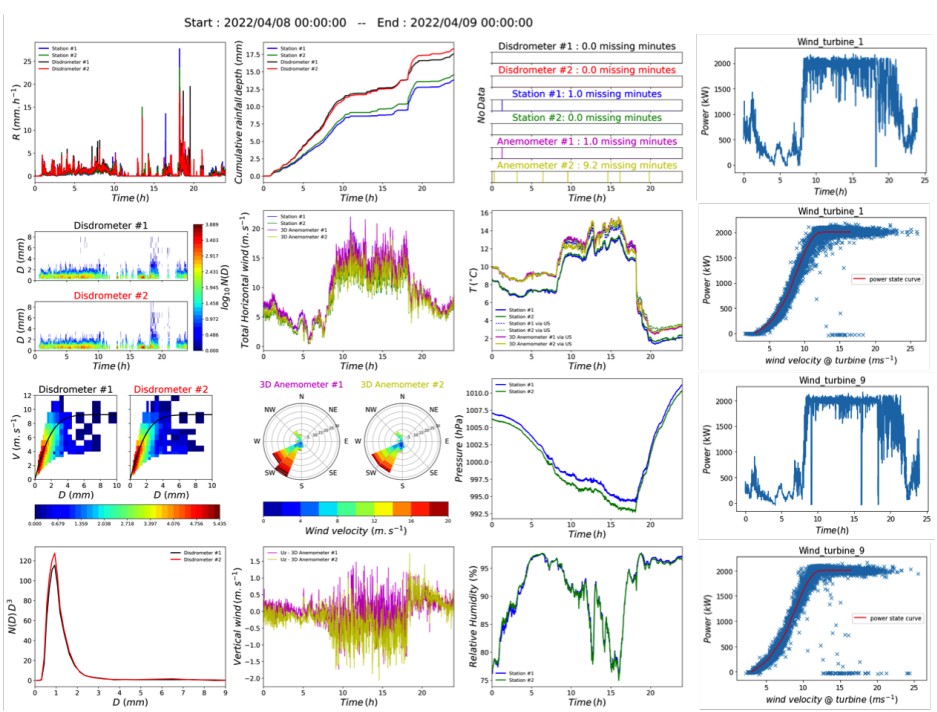

Figure 2: Quicklook of the RW-Turb data on 08 April 2022. Turbine power shown in the rightmost column is proprietary of Boralex, this is not available in the public database of RW-Turb (online quicklook). Description of the plots can be found in the text.

Technical and working information of the turbines can be found in Vestas Wind Systems A/S (2023). Vestas V90 are designed with a power configuration of 2.0 MW (rated power), pitch regulated with variable speed. The hub height of the turbines is 80 m, this is closer to the vertical height of upper set of devices on the mast (location 1 at $\approx 78$ m). The power state curves of the turbines can be seen in Fig. 2, last column; it follows the cut-in, rated and cut-out wind speeds ($4 \, \mathrm{m \, s^{-1}}$, $12 \, \mathrm{m \, s^{-1}}$ and $25 \, \mathrm{m \, s^{-1}}$) with majority of points being around the rated power. Some clustering of power values can be seen at zero, this is because of treating negative power (power consumed for operation > power produced) as zeroes in data. The power output are sampled with time steps of 15 s.





### 2.3.2. Derived fields: Wind power available and air density

Power production from turbines are analyzed at the lowest available time-step, 15 s, here (4 Vestas V90 - 2MW managed by Boralex, see Fig. 1 for location from the meteorological mast). Power available at the turbine for extraction can be approximated as:

$$P_a = \frac{1}{2}\rho A v^3 C_p \tag{10}$$

where $\rho$ is the air density at wind turbine height ($h_{hub}$), $A$ is the swept area of turbine rotor , $v$ the wind velocity (ms$^{-1}$) approximated at turbine height and $C_p$ the power coefficient or Betz coefficient (for Vestas-90 examined here, $h_{hub}$ = 80 m; A = 6,362 m$^2$, and rated power is 2 MW). A strong limitation of this widely used formula is that it does not account for the wind spatial variability over the swept area. The value of air density is often approximated as $1.255$ kgm$^{-3}$ (standard value at sea level, 15 °C). However, it is known to show fluctuations and reported to have an effect on power generation in varying levels (Jung and Schindler, 2019; Ulazia et al., 2018). For the purpose of this analysis, air density was considered as a varying quantity and estimated using the current official formula of the International Committee for Weights and Measures (CIPM), referred to as CIPM-2007 equation which accounts for humidity (Picard et al., 2008):

$$\rho(T,P,H_r) = \frac{PM_a}{Z(T,P,H_r)RT(K)}\left\{1 - x_v(T,P,H_r)\left[1 - \frac{M_v}{M_a}\right]\right\} \tag{11}$$

where $T$ (°C), $P$ (Pa) and $H_r$ ($0 \leq H_r \leq 1$) are temperature, pressure and humidity from Meteorological station at $h_{hub}$. Other derived parameters are

$T(K)$, air temperature (in K; from $T$)

$Z$, compressibility factor (a function of $T$ and $P$)

$R$, molar gas constant (J mol$^{-1}$ K$^{-1}$)

$x_v$, mole fraction of water vapour

$M_a$, molar mass of dry air (g mol$^{-1}$)

$M_v$, molar mass of water (g mol$^{-1}$)

### 2.3.3. Sampling resolution, biases and filtering of data

As it can be seen in the turbine power state curves in Fig. 2 (last column), vast majority of the turbine power ($P_t$) values are clustered around the rated value of 2.0 MW. However, when the available power ($P_a$) is calculated using the Eq. 10, the values go far beyond the limitation of rated power. This upper limit, along with the presence of zeroes was found to bias the UM estimates of Turbine power. This is addressed in part 1 of the paper, and since it was possible to retrieve those biased values from the underlying field ($P_a$) by artificially imposing the biases, it was decided to use $P_a$ as the field to study for realistic correlation values. In the analysis presented, $P_t$ is also included, however, it should be considered with the biases detected for which no corrections are available so far.





Other than this bias from rated power in turbine, there were few more concerns regarding the quality of remaining data. On the basis of data presented in Gires et al. (2022), UM analysis of the fields revealed that even though data is recorded at finer resolution, the actual sampling resolution for studying variability may be coarser. Based on this insight, the fields are analyzed here at lower resolutions than manufacturer claims (which are still

high-resolution as far as data is concerned). Table 1 summarizes the fields studied and their actual sampling resolution. This is applicable for instruments at location 1 as well as location 2 on the mast (refer Fig. 1).

| Field | Data source | measured/derived | recording resolution | actual sampling resolution |
|---|---|---|---|---|
| Temperature ($T$) | | measured | 1 Hz | 15 s |
| Pressure ($P$) | | measured | 1 Hz | 15 s |
| RH ($RH$) | Meteorological station | measured | 1 Hz | 15 s |
| Air density ($\rho$) | | derived, CIPM-2007 | 1 Hz | 15 s |
| Power available ($P_a$) | | derived ($\rho, v$) | 1 Hz | 15 s |
| Wind velocity ($v$) | 3D sonic anemometer | measured | 100 Hz | 1 Hz |
| Power produced ($P_t$) | Wind turbine | measured | 15 s | 15 s |
| wind velocity ($v_t$) | | measured | 15 s | 15 s |
| rainfall($R$) | Disdrometer | measured | 30 s | 30 s |

Table 1: Details of fields studied, their source and actual sampling resolution at which they were studied (based on results from Gires et al. (2022)). Station parameters were taken at 15 s (instead of 16 s) to match wind turbine power measurements.

Before proceeding to analysis, the whole data set was validated (Nov 2020 to May 2022) by checking for unusual entries and instrument downtimes at both locations on the

270 mast as well as 4 turbines. Time steps were not considered for all fields if any one of the devices was not working. This included 5 months when anemometer (17 June 2021 to 29 Nov 2021) and station (17 June 2021 to 11 Nov 2021) at location 1 on the mast were struck by lightning and had to be replaced, and some time steps of turbine downtime (which were given as interpolation in unfiltered data) during March and June 2021. There were few

time steps where abnormal values were recorded for $T$, $P$ and $RH$; these were removed by a simple filter that replaced values of station parameters with 'nan' (not a number) whenever pressure was shown below 800 hPa. If 'nan' were isolated, they were replaced by the average of preceding and succeeding entries.

An event was considered strictly rain, if there was a cumulative depth greater than

280 0.5 mm and separated by at least 15 minutes of dry condition before and after. The converse of this criteria was employed for getting dry events; events smaller than 5 min were





discarded as well as events when any of the devices (including turbines) are giving more
than 30 % 'nan' or 50 % zeroes. After data filtering, a total of 1488 rain events (and 2309
dry) were obtained; events were identified from 2 years and 3 month-long data (12 Nov
2020 to 09 Feb 2023). Further removal of events was performed in subsequent UM analy-
ses to accommodate event size to the closest power of 2.

### 3. Multifractal analysis of the fields

One major interest of this campaign involving simultaneous measurement of wind and
rain was to study the correlations between them across various scales. In this section, the
validity of multifractal characterization of the fields is tested using the framework of UM;
this is followed by correlated multifractal analysis using the framework of joint multifrac-
tals (JMF).

#### 3.1. UM analysis of fields according to dry and rain conditions

Before performing joint analysis, the fields were individually studied for possible dif-
ferences in behaviour during rain and dry conditions using UM analysis. Rain and dry
events were selected following the criteria mentioned in previous section, and each of the
fields in Tab. 1 were subjected to multifractal analysis for the selected events separately
as well as as an ensemble (rain ensemble and dry ensemble). Out of the events identified
using the criteria mentioned before, the events with more than 30 % of nan/zero were re-
moved by checking the data across all devices; this left 765 rain events (and 1203 dry). To
reduce the influence of upper and lower thresholds in turbine power, a further correction
was employed where columns with more than 30 % nan/zero were removed equally across
all ensembles. For UM analysis, a sample size ($N_{sam}$) of 128 (32 min) was used for fields
at 15 s and 2048 ($\approx$ 32 min) for fields at 1 Hz. If an event was larger than the sample size
(powers of 2 greater than $N_{sam}$), it was split into ensembles of length $N_{sam}$. For example, if
the length of an event is 300 (75 min), it is trimmed to the nearest power of 2 (256, 64 min)
so as to accommodate the time steps that give the largest rainfall cumulative depth; this was
then made into an ensemble of size 128 (32 min) with 2 columns. To maximize the number
of events in the analysis, events with length $< N_{sam}$ but $\geq 80\%$ of $N_{sam}$ (or powers of 2 $>$
$N_{sam}$) were included by extending their length to $N_{sam}$ (or powers of 2 $> N_{sam}$) from the
data set.

Results of an ensemble analysis of all rain events are shown in Fig. 3 (fields at 15 s) and
Fig. 4 (fields at 1 Hz). Wind velocity ($v$) was estimated as the horizontal resultant from $u_x$
and $u_y$ provided by 3D sonic anemometer; Power available $P_a$ was derived from this using
Eq. 10. Both quantities were initially estimated at an instrument resolution of 1 Hz (Fig.3)
and also averaged to 15 s (Fig. 4). Since air density ($\rho$) involves station parameters (at
15 s), the finest time step was limited by them to 15 s, which anyway corresponds to the
time step of power production available. For illustration purposes only Turbine 1 (turbine



closest to the mast, Fig. 1) is shown; other turbines gave similar estimates. The rest of the
fields were taken from instruments at location 1 of the mast ($\approx 80\,\mathrm{m}$ height) which is on a
similar horizontal plane as turbine hubs.



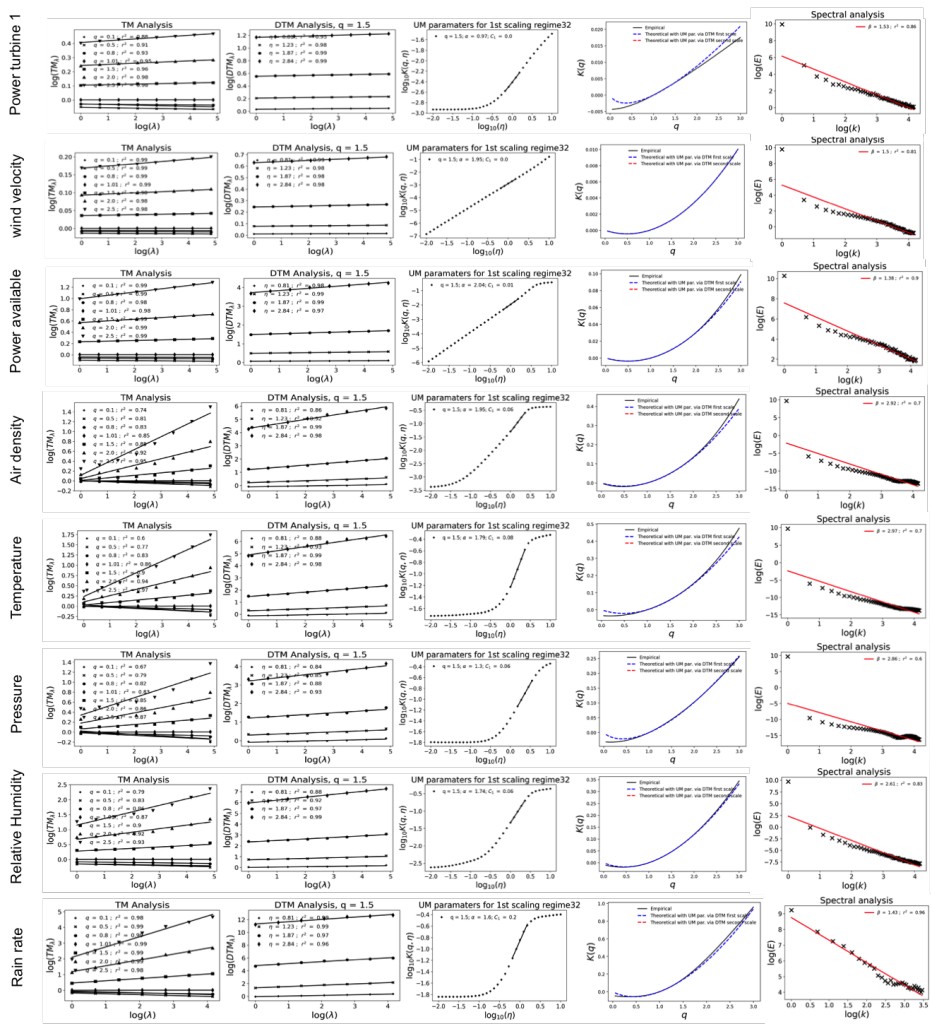

Figure 3: UM plots of rain events from 11 Dec 2020 to 03 June 2021 (6 months) for all fields studied at the lowest instrumental resolution of 15 s (except for Rain rate at 30 s). Ensemble of 756 events at a sample size of 128 (32 min), fluctuations of the field were used for station fields while direct field for rest; spectral plots here are from direct data.



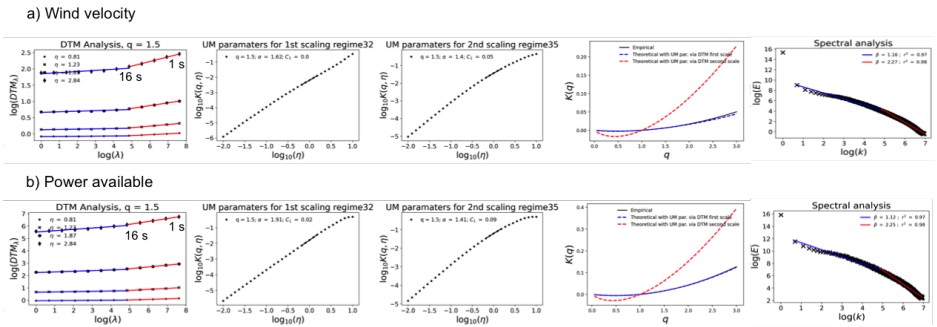

Figure 4: UM plots of rain events from 12 Nov 2020 to 09 Feb 2023 ( 2 years 3 months) for a) wind velocity and b) power available studied at the lowest instrumental resolution of 1 Hz. Ensemble of 213 events at a sample size of 2048 ($\approx 32\,\text{min}$); $\alpha$ was estimated from the slope of DTM curve at $\eta = 0$. FIF of the field was used; spectral plots here are from direct data.

UM plots for each field as an ensemble of all rain events are given in Fig. 3 and Fig. 4 for the time period considered (corresponding plots of dry events are given in appendix: Fig. A1 and Fig. A2) . The value of the non-conservation parameter $H$ was too high for
UM analysis of station fields directly - $T$, $P$, $RH$, and $\rho$ - ($H \sim 0.9$ and $\beta \sim 2.8$); this was reduced along with spectral slope to conservative values by considering the fluctuations of the fields, which is a common approximation for fractional differentiation ($H \sim 0$ and $\beta < 1$). They all gave similar $C_1$ values ($\sim 0.06$); $\rho$ and $RH$ gave similar $\alpha$ values ($\sim 1.7$) as well while $P$ and $T$ gave values of 1.39 and 1.2. For $P_a$ and $v$, the 1 Hz data, two
scaling regimes were observed with a break closer to 15 s (16 s in actuality, Fig. 4). Direct data gave estimates of $H$ acceptable ($H < 0.5$) for performing UM analysis when 15 s was used as the finest time step (Fig. 3: $H \sim 0.2$ and $\beta \sim 1.4$), while the smaller scale (1 Hz to 15 s) gave very non-conservative values ($H \sim 0.6$ and $\beta \sim 2.2$). For $P_a$ and $v$ at 1 Hz (1 Hz to 15 s), taking the fluctuations reduced H too much ($\sim$ -0.2 and -0.4 respectively). In
examining these smaller scale variations, fractionally integrated flux (FIF) is recommended for retrieving the conservative part, this gave $H \sim 0$ (Fitton, 2013; Gago et al., 2022). For $P_a$ and $v$, the values of $\alpha$ and $C_1$ were 1.91 and 0.021, and 1.62 and 0.0093 for larger scales (from 16 s to 32 min); for finer scales (1 Hz to 16 s) $\alpha$ values were smaller while $C_1$ larger : 1.40 & 0.09, and 1.38 & 0.05. The possibility of 2 scaling regimes for 15 s fields is not
considered here (Fig. 4) as it was convenient to compare rain and dry conditions in a single regime for consistency.


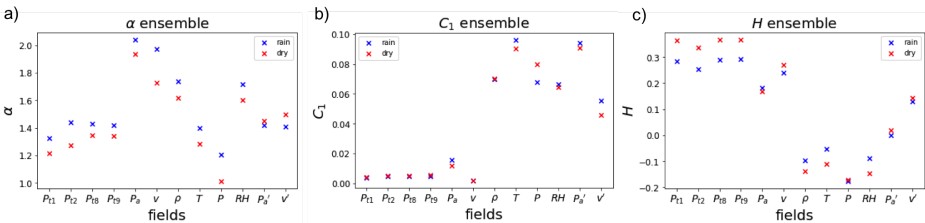

Figure 5: Comparison between UM parameters of rain and dry events ensemble: a) $\alpha$, b) $C_1$, and c) $H$.

From ensemble analysis, slightly increased values of $\alpha$ were observed for the rain ensemble in comparison to the dry ensemble (plots shown in Fig. 5) for all fields. Since $C_1$ is rather similar, it can be inferred that the fields exhibit more variability when rain is present (Fig. 5a and Fig. 5b). With this insight, rain events are analyzed in detail individually.

### 3.2. Joint analysis of fields according to rain

The scaling and multifractal properties of fields were examined for rain (and dry) events individually and as an ensemble previously. The inter influence of some of these fields are obvious by virtue of definition: available wind (and hence power extracted by turbines, $P_t$) and air density ($\rho$) are derived from wind velocity ($v$) and station fields ($T$, $P$, and $RH$) respectively. For understanding the influence of rain on wind power, it is essential to understand its natural correlation with wind (and hence power available, $P_a$). Using the previously defined framework of joint multifractals (JMF), it is possible to analyze two conservative fields together and to estimate the correlation exponent between each other when one is expressed as a multiplicative combination of the other with an independent multifractal field. For example, the correlation of $P_a$ with $v$ can be explored by expressing them as $P_{a\lambda} = \frac{v_\lambda{}^a Y_\lambda{}^b}{\langle v_\lambda{}^a Y_\lambda{}^b \rangle}$, where $\lambda$ is the resolution of the field, $Y_\lambda$ another UM field and $a$ and $b$ are the exponents of correlation between them (see section ??). .


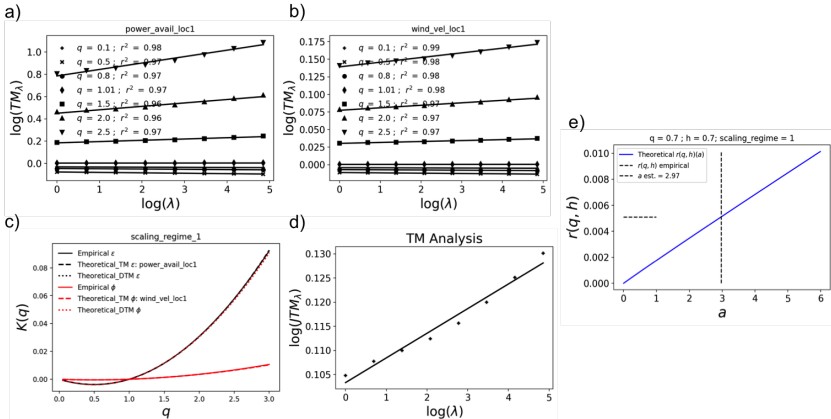

Figure 6: a) TM plots of $P_a$ , b) TM plots of $v$ (log - log plots of Eq. 3), c) K(q) plots for both fields, d) TM plot for the joint field, e) estimation of JMF parameter $a$; for an ensemble of all moderate rain events at location 1. Rain events were analyzed as an ensemble of size 128, from 12 Nov 2020 to 09 Feb 2023 ($\sim$ 2 years 3 months).

With this framework, the correlation of $P_t$, $P_a$, $v$, and $\rho$ with each other (and with station fields) are explored here according to rain rates. For this purpose, the rain events (12 Nov 2020 to 09 Feb 2023) were classified into 6 groups based on the rain rate (with 5 min moving average so that events are characterized by their intense portion). For this, the criteria used in Tokay and Short (1996) was used (only rain rate), rain events were selected and separate ensembles ($N_{sam}$ of 128 time steps or 32 minutes) were created for each of the 6 rain groups. Since JMF involves expressing fields as a combination of each other, the finest resolution of fields were limited by the highest actual sampling resolution (15 s, Table 1). JMF plots of $P_a$ and $v$ for an ensemble of all moderate rain events at location 1 are shown in Fig. 6 as an illustration for pedagogical purposes. Value of $a$ closer to 3 (as expected from Eq. 10) and good scaling was obtained with $r^2_{JMF}$ value of 0.98. The variation of JMF parameters $a$ and $IC$ are given in Fig. 7 for location 1; similar estimates were obtained for location 2 as well. Overall, a very small increase in values of $IC$ and $a$ were observed with an increase in rain rate (5 min moving average) when correlations of $P_a$ against $v$ and station fields were considered (Fig. 7b). A similar trend was observed when $v$ was analyzed against $P_a$ and station fields (Fig. 7c), and also when $\rho$ was analyzed against the rest of the station fields (Fig. 7d). Quality of scaling $r^2_{JMF}$ didn't show any trend like the values of $a$ or $IC$. The effect of the previously mentioned thresholds in turbine power (due to rated power and negative power) seems to have a stronger bias in JMF; JMF of $P_t$ with every field across various rain types gave estimates far lower than that of $P_a$ with comparatively worse scaling. The estimates were found to be even lower when the 30 %




correction was not employed (values of $a$ close to 0); without the correction, $P_t$ also gave inconsistent values of $r^2_{JMF}$ with values going lower than 0.1 in some cases. This behaviour was consistent across all four turbines. For $P_t$ (Fig. 7a), this poor scaling is not surprising, considering the biases established earlier. As a result, the interpretation of trends is not advisable and its better not to consider values of JMF parameters from $P_t$ as they are not

robust enough.

Here, values were estimated at $q = h = 0.7$ based on sensitivity analysis around various $q/h$ options (for both individual and ensemble analysis). Values of $q_s$ and $q_D$ (moment corresponding to sampling limitation and divergence respectively) were above $ha + q$, $ha$ and $q$ for all the cases analyzed here as desired, this is required for obtaining reliable values

in JMF.

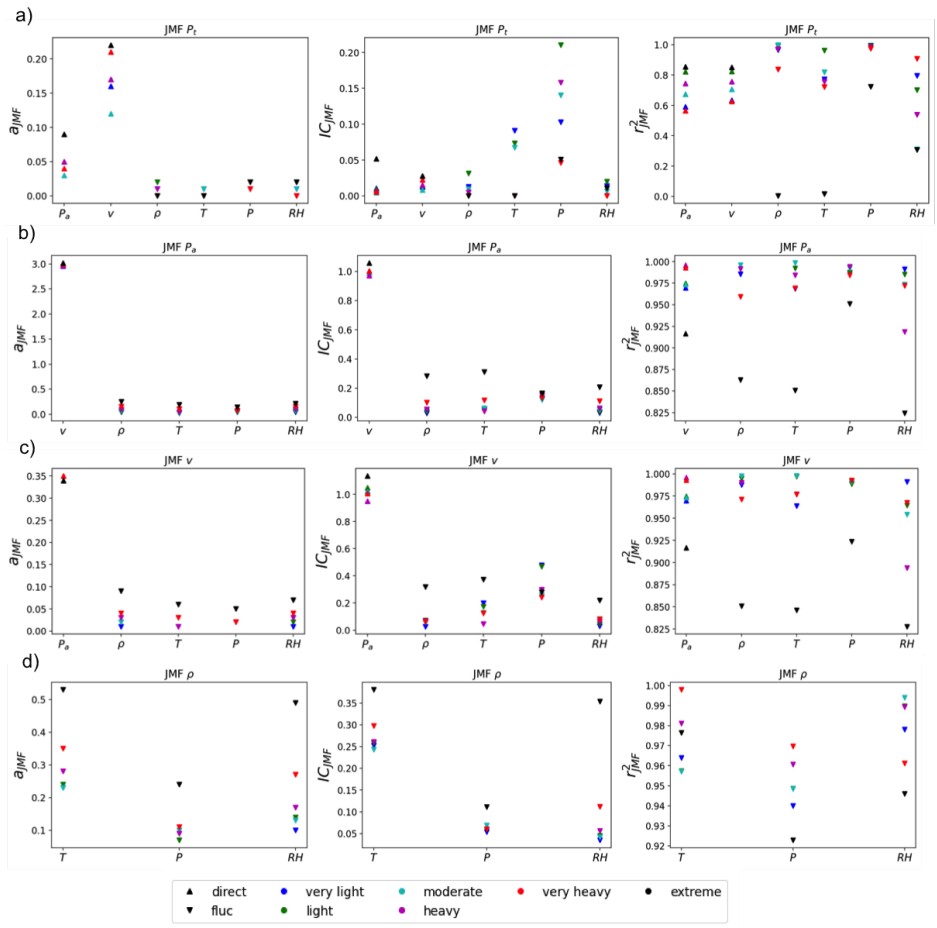

Figure 7: Variation of JMF parameters $a$, indicator of correlation $IC$, and quality of scaling $r^2_{JMF}$ between a) $P_t$ (Turbine 1), b) $P_a$, c) $v$, and d) $\rho$ and other fields according to type of rain (on the basis of 5 min moving average of rain rate with criteria in Tokay and Short, 1996). Rain events of each class were analyzed as an ensemble of size 128, from 12 Nov 2020 to 09 Feb 2023 ($\sim$ 2 years 3 months).

From early UM analysis, it was decided that for fields at 15 s resolution, all station fields need to be analyzed as fluctuations while wind ($v$) and wind-derived fields ($P_a$ and $P_t$) can be studied directly. Though the desired conservative field is retrieved by this choice, this could cause issues in JMF as it could be a combination of a direct field and an indirect field



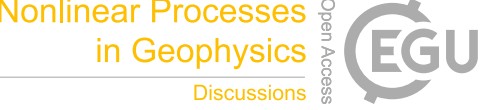

(fluctuations or FIF). For example, in Fig. 7a, $P_a$ is a direct field while the fields with which
its correlations are analyzed ($\rho$, $T$, $P$, and $RH$) are fluctuations. In UM this is discussed
in the previously defined Eq. 5. To recap, a non-conservative field $\psi_\lambda$ (i.e. $\langle \lambda' \rangle \neq 1$); in
UM, can be expressed in terms of the underlying conservative field ($\varepsilon_\lambda$ retrieved through
fluctuations or FIF, $\langle \varepsilon_\lambda \rangle = 1$) as $\psi_\lambda = \varepsilon_\lambda \lambda^{-H_\varepsilon}$. Here $H_\varepsilon$ is the non-conservation parameter

that characterizes the variation of the mean of $\varepsilon_\lambda$ across resolutions $\lambda$. When two fields $\varepsilon_\lambda'$
and $\phi_\lambda'$ ($'$ to suggest non-conservative nature) are analyzed as a multiplicative combination
in JMF, only their respective conservative parts can be used ($\varepsilon_\lambda = \frac{\phi_\lambda{}^a Y_\lambda{}^b}{\langle \phi_\lambda{}^a Y_\lambda{}^b \rangle}$). Hence, the
estimated JMF parameter $a$ doesn't correspond to the full field. If one field is direct and
the other is a retrieved conservative part (fluctuations or FIF), values of $a$ could be biased

as underlying $H$ ($H_\varepsilon$ and $H_\phi$) is not considered in its estimation.

To assess the possible influences of this, a sensitivity analysis was performed using
two known fields: $P_a$ ($\varepsilon_\lambda$) and the field it is derived from $v$ ($\phi_\lambda$): $P_a \propto v^3$ (Eq. 10). The
previously used dataset - respective ensembles of rain events from 12 Dec 2020 to 03 June
2021 (6 months, with $N_{sam}$ 128) - was used for this purpose; the results are displayed in

Table. 2. While using $P_a$ and $v$ as direct fields, $a$ in JMF analysis retrieved the exponent
value in Eq.10 (Table.2) with good joint scaling ($r^2_{JMF}$) and indicator value ($IC$). Though $H$
isn't non-zero for either of the fields, they being similar gave a difference close to zero ($H_\varepsilon$
- $H_\phi$). Similarly, a closer value of $a$ ($a = 2.75$) was obtained when both fields were taken as
FIF. From the samples in Fig. 8a and Fig. 8b, it can be seen that the fields follow the same

pattern when both fields are direct or FIF (Fig. 8b follows the same pattern as direct field
in Fig. 8a while fluctuations in Fig. 8c does not) with the difference in amplitude from
the mean line following the proportionality exponent in Eq. 10. When both fields were
taken as fluctuations, values of $a$ closer to 1 were obtained. This is rather consistent as
fluctuations take the difference between time steps and are expected to show a proportional

relationship as the fields are already related. However, this also puts the analysis at an
apparent disadvantage as using JMF on fluctuations only retains the proportionality but not
its order. This can be observed in the sample in Fig. 8c, where both fields appear moreover
similar (following $P \propto v$ than the original $P \propto v^3$). In the remaining cases, - when both
fields were not having similar values of $H$ - the estimates of $a$ are decreased except when

$H_\varepsilon$ was significantly lesser than $H_\phi$ (FIF - $P_a$ and direct - $v$). This might have to do with
$\varepsilon_\lambda$ ($P_a$) being the field estimated based on $\phi_\lambda$ or $v$ ($P_a = \frac{v^a Y_\lambda{}^b}{\langle v^a Y_\lambda{}^b \rangle}$) while the JMF analysis is
trying to express it in terms of fluctuations of $\phi_\lambda$ which doesn't follow the same time step
pattern as direct data or FIF (Fig. 8c).


| $\varepsilon_\lambda$ | $\phi_\lambda$ | $H_\varepsilon$ | $H_\phi$ | $H_\varepsilon - H_\phi$ | $a$ | $b$ | $IC$ | $r^2_{JMF}$ |
|---|---|---|---|---|---|---|---|---|
| | direct | 0,210 | 0,256 | **-0,045** | **2,98** | 0,823 | 0,993 | 0,994 |
| direct | FIF | 0,210 | -0,026 | 0,237 | 1,62 | 0,696 | 0,895 | 0,953 |
| | fluc | 0,210 | -0,253 | 0,464 | 0,02 | 0,537 | 0,012 | 0,430 |
| | direct | -0,004 | 0,256 | -0,259 | 4,57 | 0,843 | 0,934 | 0,960 |
| FIF | FIF | -0,004 | -0,026 | **0,022** | **2,75** | 1,179 | 0,990 | 0,888 |
| | fluc | -0,004 | -0,253 | 0,250 | 0,01 | 0,806 | 0,002 | 0,043 |
| | direct | -0,182 | 0,256 | -0,438 | 1,7 | 9,965 | 0,082 | 0,956 |
| fluc | FIF | -0,182 | -0,026 | -0,156 | 0,73 | 4,729 | 0,049 | 0,973 |
| | fluc | -0,182 | -0,253 | **0,071** | **1,01** | 0,397 | 0,892 | 0,779 |

Table 2: Sensitivity analysis using power available, $P_a$, ($\varepsilon_\lambda$) and wind velocity, $v$ ($\phi_\lambda$) where JMF parameters are estimated for different combinations of data - direct (dir), fluctuations (fluc), and FIF (fractionally integrated flux). Data from 12 Dec 2020 to 03 June 2021 at 15 s, fields were renormalized for comparison.

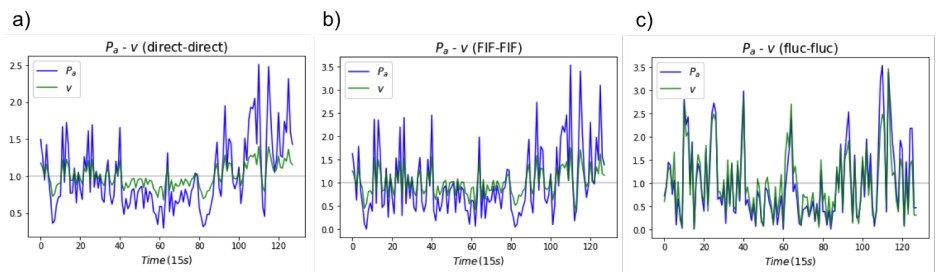

Figure 8: a) direct data of $P_a$ and $v$ b) FIF of $P_a$ and $v$, and c) fluctuations of $P_a$ and $v$ for one sample ($N_{sam} = 128$) of the data analyzed (from 12 Dec 2020 to 03 June 2021 at 15 s, fields renormalized for comparison). Between plots, it can be seen that direct and FIF are following similar data pattern while fluctuations does not.

Though the biases from the analysis of JMF are acknowledged here, there is no correction available at this point and this should be further investigated in future. Among the results presented in Fig. 7, all JMF analysis except for $P_a$ - $v$ combinations are affected by this. More research is needed to account for this in the framework when accurate retrieval of correlation parameters is of interest. Even with biases, the values of $a$ and $IC$ are still strong indicators for comparing two multifractal fields, through various atmospheric conditions as illustrated in Fig. 8.




## 4. Effect of rain type and wind direction in power production

### 4.1. Possible influence from convective and stratiform rain

The yearly average cumulative depth of rain at the wind farm was found to be $\sim 600$ mm and among the 6 months of rain events (213) studied, only 20 could be classified as heavier rainfall events (heavy, very heavy, and extreme). Because of this, it was speculated that the lack of a very strong correlation between rain and power produced could be due to rainfall events being not strong enough (apart from the known bias from threshold due to rated power). To test this hypothesis, efforts were made to identify the rain events as convective and stratiform. While convective rains have highly concentrated intensities, stratiform rains are more horizontally spread with lower intensities (Houze Jr, 2014; Marzano et al., 2010). Several criteria have been used for detecting this indirectly in literature; simple ones are the classification on the basis of rain rate exceeding a particular value. Popularly used criteria using rain rate is by Bringi et al. (2003) where convective rain samples are considered as those with rain rate, $R, \geq 5\,\mathrm{mmh}^{-1}$ and standard deviation (std dev) over 5 consecutive 2-minute samples $> 1.5\,\mathrm{mmh}^{-1}$ (mentioned as BR03 from here on). Tokay and Short (1996) proposed an empirical classification based on DSD parameters by identifying the shift from spectra dominated by small to medium drops (stratiform) to spectra dominated by large drops (convective) for similar rain rate (mentioned as TS96 from here on). Attributing temporal shift in DSD parameters (shape parameter $\Lambda$) to shifts in rainfall size distribution, they suggested a value of $\Lambda = 17R^{-0.37}$ above which precipitation can be considered as convective (stratiform if below).

To explore this, DSD parameters of rain events at the wind farm were estimated assuming a gamma distribution (following the method of moments used in Jose et al. (2022)). From 12 Nov 2020 to 09 Feb 2023, from the filtered list of events, a total of 150 were identified as convective (using TS96 criteria). However, only 37 events were above 32 minutes and hence among the events subjected to UM and JMF analysis before. 25 events of comparable length were selected from both convective and stratiform sides where at least 70 % of the time steps followed TS96 criteria. Two turbines were examined for these events - Turbine 1 and 9 (closest and farthest to the mast): possible difference in turbine power between convective and stratiform events is not obvious from mean - standard deviation nor state curves (Fig. 9). This obviously comes with the disclaimer that it was a simple test using limited events without considering other complexities. For example, the dispersion being greater at Turbine 9 (as it is farthest from the mast from where velocity was measured) is ignored. However, considering the predominant stratiform nature of rain at the location studied, the hypothesis of needing stronger rainfall to see the proper correlation between power produced and rainfall is still worth exploring in the future.

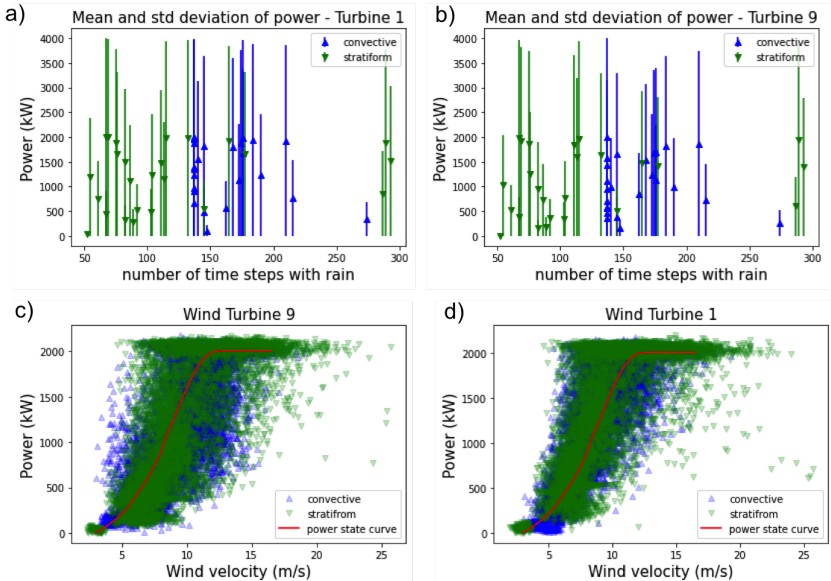

Figure 9: Mean and standard deviation of Power produced, $P_t$, for a) Turbine 1 (closest to the mast), and b) Turbine 8 (farthest from the mast). Power state curve during selected convective and stratiform events for c) Turbine 1 (closest to the mast), and d) Turbine 8 (farthest from the mast).

### 4.2. Possible influence from wind direction

The turbines are aligned southeast within a 4 km radius, and at the south of the mast a small groove is located at roughly 160 m, and a larger one in the East at around 100 m (Fig. 1). To see the effect of these topographical features and spread of vegetation around the mast, wind directions were identified as shown in Fig. 10 with mast as the centre. Based on this, average wind direction was calculated for rain events using $u_x$ and $u_y$ from 3D anemometer at location 1. Based on the position of immediate vegetation around the mast, the wind zones were grouped into three - least influenced (69), most influenced (60) and turbine direction (7 events). Since, this was done manually, on the basis of vicinity and size of vegetation, not all directions are considered in this classification (specified in Fig. 10).



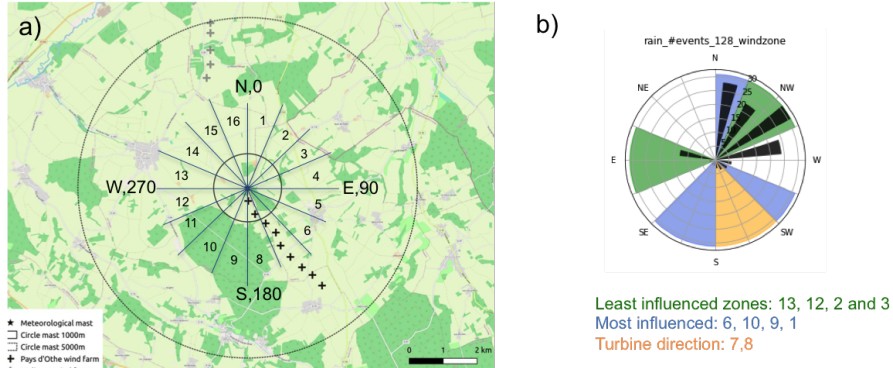

Figure 10: a) Location of wind farm and the wind directions identified, b) No. of events corresponding to the direction (colours show the direction classes, length of the black arcs corresponds to number of events while thickness to average magnitude) and the three groups considered.

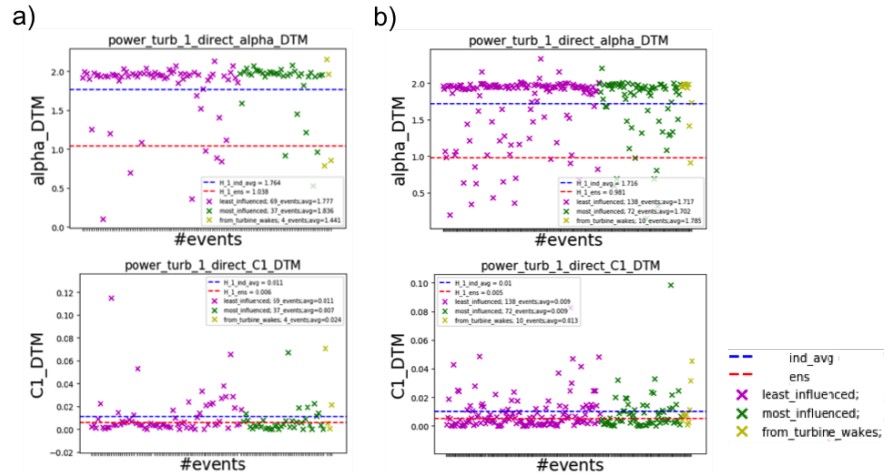

Figure 11: Variation of $\alpha$ and $C_1$ according to wind direction for a) rain events b) dry events. Values of ensemble and average value of individual events are shown using red and blue lines.

Variations of UM parameters of turbine power closest to the mast (Power turbine 1) according to wind classes are shown in Fig. 11 for rain and dry events. No obvious difference was observed, similar results were observed for rest of the turbines as well. Due to the previously identified bias from rated power in UM analysis, it is not possible to say exactly





if this is the exact behaviour or not. This was not explored further in this thesis. Factors known to affect power production at turbine wake, such as mixing of moist air Obligado et al. (2021), dynamic effects from inertial particles (Smith et al., 2021) etc. were also not considered here.

### 4.3. Power state curve at different rain and wind conditions

All the studies so far were event-focused as it provides the behaviour of a continuous field for a given period of time. In this section, instantaneous (subject to recording time step) empirical turbine power ($P_t$) was examined according to the type of rain (same criteria as those of events) and wind (14 classes at intervals of $2\,\mathrm{ms}^{-1}$). For this, all the individual time steps from 12 Nov 2020 to 09 Feb 2023 were grouped according to the $R$ and $v$ at that instant at a time step interval of $1\,\mathrm{m}$. A total of 503085 one-minute long time steps were grouped in this way. Fig. 12a shows the power curves for each rain class of Turbine 1 alongside the theoretical state curve provided by the manufacturer (dotted red line). Singular values of power were obtained by averaging all the empirical power registered at time steps corresponding to that particular wind class. This is then compared with the state curve of dry (no rain) timesteps (solid yellow line) visually (Fig. 12a and Fig. 12c), and through the percentage change: (Power$_{rain}$ - Power$_{no-rain}$) / Power$_{no-rain}$ (Fig. 12b).

At lower wind velocity classes (below $10\,\mathrm{ms}^{-1}$), the average power of all rain classes are above that of the theoretical state curve (except for 'extreme' which trails below the power curve only till $8\,\mathrm{ms}^{-1}$). Lower rain-class timesteps are generating more power in this region than heavier ones, as well as dry timesteps; this is progressively reduced as we move towards the rated wind speed (from 60% difference to almost 0% near $12\,\mathrm{ms}^{-1}$) and above. Around the rated power, state curves of all rain classes go below the state curve by manufacturer, with the difference regained as the curve moves towards cut-off speed ($25\,\mathrm{ms}^{-1}$). When compared with that of dry state curve, it can be seen that 'very light', light, and moderate rain are following closer while the rest trail below (more clearly observed in terms of percentage in Fig. 12b). It can be inferred that, there is a general increase in power produced in low rain and wind conditions, however, this behaviour is observed below the rated velocity of the Turbine. For greater winds (above rated wind velocity), the power produced during lower rains remains the same while heavier rains provide much reduced values. It can be roughly said that the heavier the rainfall, the sooner the fall of power below that expected from the state curve provided by the manufacturer.

However, this observation doesn't involve same number of $1\,\mathrm{min}$ time steps for all rain classes. For example, the 'no rain' time steps are way larger in number than rain events (Fig. 12d); in the case of rain events, higher the value of rain rate, lower the number of time steps available (Fig. 12d and Fig. 12e). To improve statistics, the events in and above 'heavy' were combined into one class; the shift observed before can be seen in this case as


well (Fig. 12c). This disproportionate number of points is also the reason for sudden dips
in the state curves at higher rain classes.

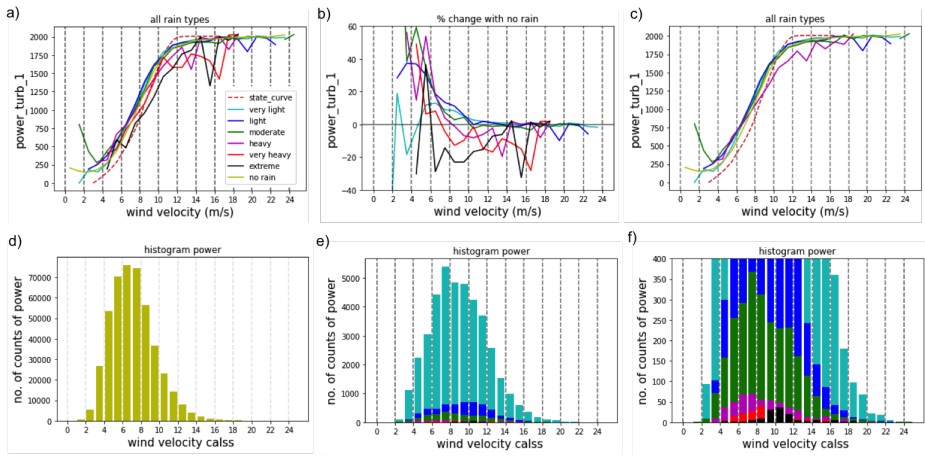

Figure 12: Power state curves by averaging power values of Turbine 1 at time steps of 1 min: a) state curves
for all rain classes, b) percentage change from state curve corresponding to time steps with no rain, c) state
curves for rain classes (rain steps in and above 'heavy' is considered as one - 'heavy'). The second column
shows histograms of time steps of different rain classes: d) 'no rain', e) all rain classes, and f) all rain classes
(zoomed for higher rain).

Fig. 13a shows the same information but at time steps of 10 min. To respect the scale
change, rain rates were grouped as per singularities ($\gamma$); rainfall singularity ($\gamma_r$) for the rates
at 1 min were used for categorizing rain rates at 10 min ($\gamma_r = \frac{log(rain\ rate)}{log(\lambda)}$). This reduces the
average rain rates to corresponding lower values.

However, this also truncates the extreme rain time steps (of 10 min) due to lack of
points; this can be seen in Fig. 13e and Fig. 13f. This is reflected in the uneven distribution
of state curves for higher rain steps. Still, as seen before (in Fig. 12a), rain below 'heavy'
are shifting from the theoretical state curve around 10 - 12 ms$^{-1}$ velocity class here as well.
This can be seen in a cleaner way in Fig 12b where higher rain time steps are combined
into one - 'heavy'. Fig. 12b shows the percentage difference of this shift with respect to
values at no rain.



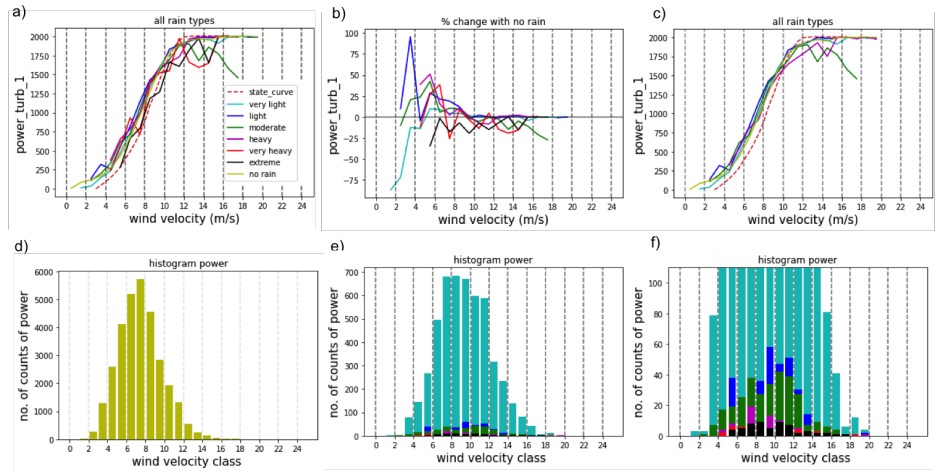

Figure 13: Power state curves by averaging power values of Turbine 1 at time steps of 10 min: a) state curves for all rain classes, b) percentage change from state curve corresponding to time steps with no rain, c) state curves for rain classes (rain steps in and above 'heavy' is considered as one - 'heavy'). Second column shows histograms of time steps of different rain classes: d) 'no rain', e) all rain classes, and f) all rain classes (zoomed for higher rain).

To summarize this observation in terms of turbine state curve values, rain steps below 'heavy' falls below the theoretical state curve after 10 - 12 ms$^{-1}$ which corresponds to the transition of power curve to rated power (12 ms$^{-1}$). It can be roughly inferred that the higher the rain rate, the lower the velocity at which the power falls below the expected value for the velocity at that time step. Also, after the cut-in velocity (4 ms$^{-1}$), heavier rains show a higher percentage difference from those of 'no rain' (Fig 12b and Fig 13b).

As different shifts from the theoretical state curve were observed for different rain classes, the JMF analysis earlier was re-performed by dividing the events on the basis of wind velocity. Since the shift happened around the rated speed of 12 ms$^{-1}$, events were grouped into '< 10' and '>10'. 10 ms$^{-1}$ was selected on the basis of the above-mentioned observations as well as by considering some leeway for the shift to rated power. The variation of JMF parameter 'a' is shown in Fig. 14. The trend observed in Fig. 7 is mostly lost here since splitting the events on the basis of velocity reduced the number of data sets available for analysis, esp for higher rain events. Further, the biases associated with empirical power ($P_t$) make meaningful interpretation difficult. It is not possible to characterize the behaviour observed from time step based analysis on events with current data and methodology. Further, since the velocity is averaged over larger time periods when it comes to

events the information is diluted to some extent as well.

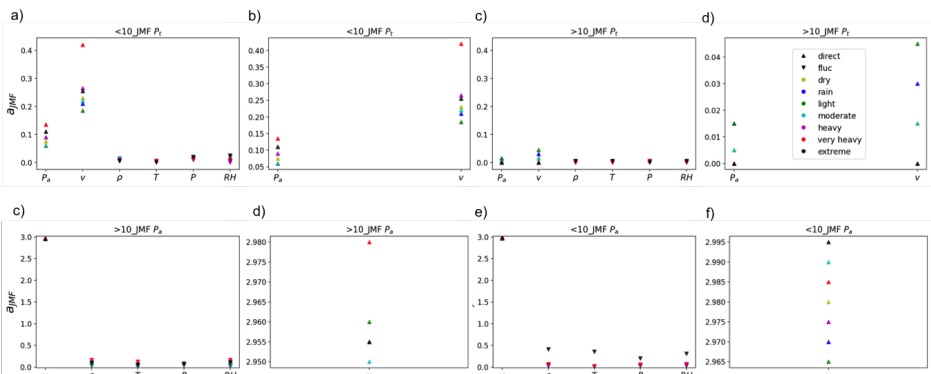

Figure 14: Variation of JMF parameters 'a' for an ensemble of events (dry and various rain classes) at a
sample length of 32 min. First row for events with average wind velocity $< 10\,\mathrm{ms}^{-1}$ a) Pt (Turbine 1), b) Pa,
c) v, and d) and other fields according to type of rain (on the basis of 5 min moving average of rain rate with
criteria in Tokay and Short, 1996). Rain events of each class were analyzed as an ensemble of size 128, from
12 Nov 2020 to 09 Feb 2023 ( 2 years 3 months)

## 5. Conclusion

From Gires et al. (2022), it was identified that the actual sampling resolution relevant
for studying the variability of meteorological fields measured with the help of mini-station
(temperature $T$, pressure $P$, humidity $RH$, and air density $\rho = f(T,P,RH)$) and that for
3D anemometer fields (wind velocity $v$, power available $P_a$) were 15 s and 1 s respectively
(instead of 1 s and 0.01 s). Using the data averaged to these reliable frequencies, UM be-
haviour, as well as JMF correlation between $P_t$, $P_a$, $v$, $\rho$, $T$, and $RH$, were analyzed to gain
insights into its correlation with rainfall, which is poorly understood. However, the direct
analysis of turbine power was found to be difficult since the output from wind turbines is
limited by a maximum or rated power; in time series analysis this acts as an upper threshold
resulting in reduced estimates of UM parameters. This bias is identified in the theoretical
framework of UM and is also illustrated using discrete cascades numerical simulations of
conservative multifractal fields in part 1 of this joint paper. Due to the presence of these
biases in $P_t$, the actual wind power available at the turbine hub for extraction ($P_a = f(v,\rho)$)
was primarily used instead as the main field for joint analysis.

For UM analysis, fluctuations of the fields were required for station fields, for retrieving
conservative fields so that estimates of TM and DTM are not biased. For anemometer fields,
direct field analysis was acceptable in large-scale regimes (from 15 s) while small scales




(0.01 s to 15 s) required retrieval of conservative fields through FIF. From UM analysis of rain and dry events as ensembles, it was found that almost all fields are showing a slight
increase in variability with rain (larger $\alpha$ and similar $C_1$) in the scale range from 15 s to 32 min, over which a unique scaling behavior is identified. An opposite trend was observed for finer scales of $P_a$ and $v$ (0.01 s to 15 s). Joint analysis of $P_a$, $v$ and $\rho$ against each other and with station meteorological fields (all fields at 15 s) revealed an increasing trend in the value of JMF correlation exponent $a$ and $IC$ with rain rate. However, this is not
without biases since station fields were fluctuations while anemometer fields were direct in the analyzed scaling regime. The influence of this bias is identified and commented on. Also, detailed sensitivity analyses were made to identify the possible effects of wind direction and rainfall type on power production in turbines. No clear trends in the results were identified. Grouping of instantaneous time steps of power according to velocity and
rain revealed interesting departure from state curves for different rain classes. At lower velocities (below rated power) and lighter rains, the turbines provided power more than expected of their theoretical state curve. At higher velocities, lighter rain timesteps more or less provided expected values of empirical power while those of heavier rains provided much less. However, it was not possible to identify this on an event-basis analysis in the
current study (Fig. 14).

Future methodological developments in JMF framework are proposed here for handling the biases in analyzing direct and non direct fields. Though the effect of the upper threshold is identified in the framework, further work is required to precisely quantify the bias. Also, considering the predominant stratiform nature of rain at the measurement location, study-
ing the correlations under convective conditions is encouraged, for the future, to expand the understanding on correlations between rainfall and wind power production. The trend observed with power state curve needs more careful future examination as well. The results here are from instantaneous data analysis, this needs to be complemented with simulations and a better understanding of the physical process leading to this. Though the changes
in atmospheric conditions are considered here to some extent, the effects due to physical nature of the blade (weight, roughness etc.) and its aerodynamic interaction in flow etc. are missing.

**Appendix:**

*UM plots of dry event as an ensemble for all fields*

UM plots of dry events at RW-Turb mast, location 1: all fields at resolution of 15 s



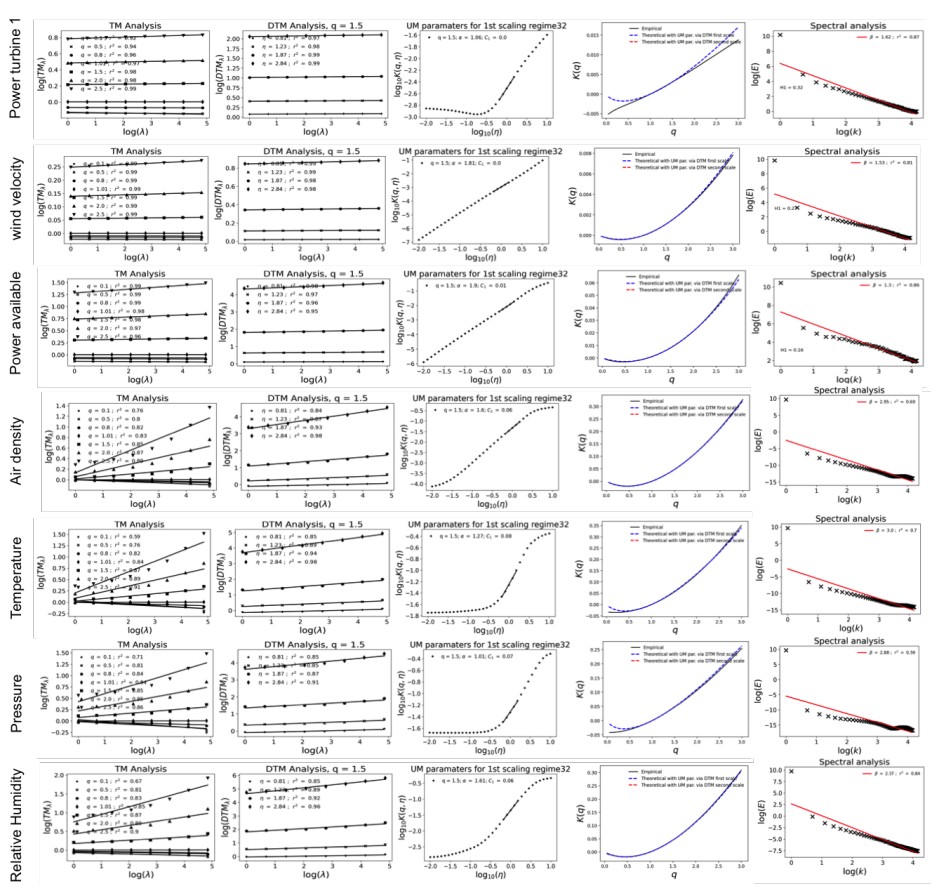

Figure A1: UM plots of rain events from 11 Dec 2020 to 03 June 2021 (6 months) for all fields studied at the lowest instrumental resolution of 15 s (except for Rain rate at 30 s). Ensemble of 213 events at a sample size of 128 (32 min), fluctuations of the field were used for station fields while direct field for rest; spectral plots here are from direct data.

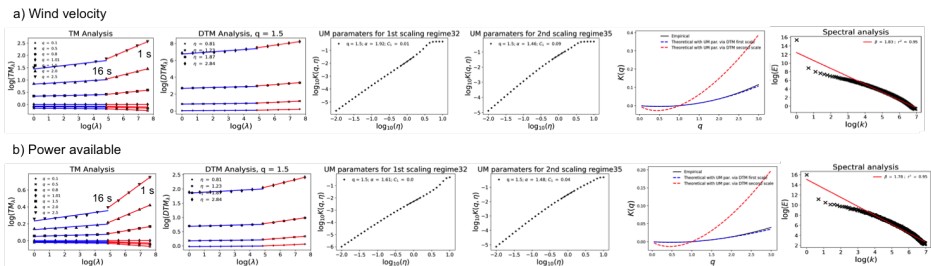

Figure A2: UM plots of dry events from 11 Dec 2020 to 03 June 2021 (6 months) for a) wind velocity and b) power available studied at the lowest instrumental resolution of 1 Hz. Ensemble of 213 events at a sample size of 2048 ($\approx$ 32 min); $\alpha$ was estimated from the slope of DTM curve at $\eta = 0$. FIF of the field was used; spectral plots here are from direct data.

## Competing interests

At least one of the (co-)authors is a member of the editorial board of Nonlinear Processes in Geophysics.

## Acknowledgement

The authors greatly acknowledge partial financial support from the Chair of Hydrology for Resilient Cities (endowed by Veolia) of the École des Ponts ParisTech, EU NEW INTERREG IV RainGain Project, EU Climate KIC Blue Green Dream project, the Île-de-France region RadX@IdF Project, and the ANR JCJC RW-Turb project (ANR-19-CE05-0022-01).





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
