# Peer review of "Part 2: Joint multifractal analysis of available wind power and rain intensity from an operational wind farm"

_Nonlinear Processes in Geophysics, 2024_

## Referee Comment (RC1)

The authors characterize the small-scale fluctuations in wind power production using data from an operational wind farm at 70 Pays d'Othe, 110 km southeast of Paris, France, and Universal Multifractals framework. The main objective of this article is to highlight differences between rain and dry conditions for the fields illustrating the influence of rain. For this purpose, the joint multifractal analysis framework and indicator of correlation (IC) was introduced and observed between various fields with an increase of *IC* in rain rate. Finally, the authors examine the possibility of difference in power production according to type of rain (convective or stratiform) as well as various regimes of wind velocity.

**Major issues**

1. The abstract of the article is too long, around 29 lines. The authors should be more concise in the abstract because several of their ideas would be better in the introduction.

2. Eq. 5 presents the multifractal behavior for a non-conservative field with parameter H. Also, it is known that other important measures in multifractality are the Renyi entropy or the generalized fractal dimension (see https://doi.org/10.1088/1361-6633/ab42fb). Therefore, there remain an important point to be addressed in this direction and that should be mentioned in the article to establish future work directions of this article: What is the relationship of the parameter H with other multifractal measures such as the Hurst exponent generalized or the generalized fractal dimension?

**Minor issues**

1. Eq. (2), (3), (4), (5) and (6), do not have explicit references from which they were taken before being placed as was done for Eq. (1). The above, although it is a minor change, is suggested so that those readers who do not know much about the Universal Multifractals approach can inquire about it, and therefore, for the article to have a greater scope.

2. Figures 3 and 4 need a higher resolution; when zooming in on them, some legends or information are not visible.

3. In section 3.2, line 358, there is a missing reference: *"... the exponents of correlation between them (see section ??)."*.

4. In section 3.2, line 386, there is an undefined variable "h".

5. The authors could highlight the difference of this joint multifractal analysis with others where the partition function or cross-correlation approach are introduced in the estimation of multifractal exponents (see for example https://doi.org/10.1073/pnas.0911983106 and *Wen-Jie Xie et al 2015 New J. Phys. 17 103020*).

---

## Referee Comment (RC2)

**Manuscript number :NPG-2024-6**
**Title : Part 2: Joint multifractal analysis of available wind power and rain intensity from an operational wind farm**

*Jerry Jose, Auguste Gires, Ernani Schnorenberger, Yelva Roustan, Daniel Schertzer, Ioulia Tchiguirinskaia*

**Review comments**

This study entilted « Joint multifractal analysis of available wind power and rain intensity», investigates the quantification of the effect of rainfall on wind power through the scale invariant framework of Universel Multifractals.

This manuscript is structured as follows : after an introduction part, in the part 2 the authors describe the framework of UM and JMF. In the part 3 the results of analyses of UM and JMF are presented for respectively individual and jointly data fields. The part 4 concerns a discussion part on the influence of rain type as well as that of wind direction on power production. Section 5 concludes the study and summarizes the results.

***In this study, the authors propose a new parameter JMF, from UM framework, to quantify the effect of rain on wind power output. This is represented a novelty for the scientific community and can be interest the eolian energy scientific community. However, the power output analyzed are values of available power output, instead of actual power output due to the presence of biais as indicated by the authors. For the understanding, this would relevant to insert the reference explaining this point or add in the manuscript the corresponding simulations.***
* * *
**Minor Revisions**

I suggest to authors to zoom the following result figures n°3, 4, 11-14, A1 and A2.
In fig. 3, K(q) curve represented in dotted red line is not visible.
Typos : line page 16 line 358 (see section ??)..

---

## Author Comment (AC1)

**Response to comments from editors and reviewers**

Part 2: Joint multifractal analysis of available wind power and rain intensity from an operational wind farm (npg-2024-6)

preprint in Nonlinear Processes in Geophysics from 02 Feb 2024
Editor's decision received on
* * *
The authors would like to thank the reviewer for evaluating the paper and providing a detailed feedback. Please find below our point-by-point response to the comments (Reviewer comments are shown in black and author responses are in blue).

**Anonymous Referee #1, 13 Mar 2024**

Below I attach my evaluation of the manuscript. Although the manuscript has two major issues and given that this article is the second part or continuation of a previous work which I was also able to evaluate, I trust that the corrections in the first part make these points become minor problems in this second part of this investigation.

The authors characterize the small-scale fluctuations in wind power production using data from an operational wind farm at 70 Pays d'Othe, 110 km southeast of Paris, France, and Universal Multifractals framework. The main objective of this article is to highlight differences between rain and dry conditions for the fields illustrating the influence of rain. For this purpose, the joint multifractal analysis framework and indicator of correlation (IC) was introduced and observed between various fields with an increase of IC in rain rate. Finally, the authors examine the possibility of difference in power production according to type of rain (convective or stratiform) as well as various regimes of wind velocity.

*Major issues*

1. The abstract of the article is too long, around 29 lines. The authors should be more concise in the abstract because several of their ideas would be better in the introduction.

   Thank you, we have trimmed the abstract for better readability into 20 lines.

2. Eq. 5 presents the multifractal behavior for a non-conservative field with parameter H. Also, it is known that other important measures in multifractality are the Renyi entropy or the generalized fractal dimension (see `https://doi.org/10.`

1088/1361-6633/ab42fb). Therefore, there remain an important point to be addressed in this direction and that should be mentioned in the article to establish future work directions of this article: What is the relationship of the parameter H with other multifractal measures such as the Hurst exponent generalized or the generalized fractal dimension?

A response to this comment is included in Part 1. Copying the relevant portion here: As rightly pointed out, the exponent denoted '$H$' in the Universal Multifractal framework (like this paper) characterizes the degree of conservation of the mean field across scales ($H > 0$ specifying growth with scale and $H < 0$ decrease). However, the UM parameter $H$ is not identical to the classical Hurst exponent, which in any case has undergone a number of modifications/generalisations. But both quantify long range correlations for $H > 0$. This is clarified in the text to avoid confusion. We also emphasise that multifractality requires more than a scaling exponent to be statistically characterised, contrary to uni/mono-fractals.

*Minor issues*

1. Eq. (2), (3), (4), (5) and (6), do not have explicit references from which they were taken before being placed as was done for Eq. (1). The above, although it is a minor change, is suggested so that those readers who do not know much about the Universal Multifractals approach can inquire about it, and therefore, for the article to have a greater scope.

   Equations (2) to (6) are now updated to follow the format of Eq. (1). Same convention is applied for revised version of Part 1 as well. However, some references had to be repeated since the framework as a whole was taken from (Schertzer and Lovejoy, 1987, 1988)

2. Figures 3 and 4 need a higher resolution; when zooming in on them, some legends or information are not visible.

   These figures and subsequent dry versions in Appendix (A1 and A2) are now updated with higher resolution.

3. In section 3.2, line 358, there is a missing reference: "... the exponents of correlation between them (see section ??).".

   Thank you, this is corrected with reference to the right section.

4. In section 3.2, line 386, there is an undefined variable "h".

   Thank you, this corresponds to the moment of individual field in joint multifractal analysis. This is clarified along with a reference in text.

5. The authors could highlight the difference of this joint multifractal analysis with others where the partition function or cross-correlation approach are introduced in the estimation of multifractal exponents (see for example `https://doi.org/10.1073/pnas.0911983106` and *Wen-Jie Xie et al 2015 New J. Phys. 17 103020*).

Joint multifractals being derived from UM, an explicitly stochastic multifractal framework, is applicable to space-time fields and not only to time processes. This aspect is now added in manuscript.

**References**

Schertzer, D., Lovejoy, S., 1987. Physical modeling and analysis of rain and clouds by anisotropic scaling multiplicative processes. Journal of Geophysical Research: Atmospheres 92, 9693–9714. URL: `https://agupubs.onlinelibrary.wiley.com/doi/abs/10.1029/JD092iD08p09693`, doi:10.1029/JD092iD08p09693.

Schertzer, D., Lovejoy, S., 1988. Multifractal simulations and analysis of clouds by multiplicative processes. Atmospheric Research 21, 337–361. URL: `http://www.sciencedirect.com/science/article/pii/016980958890035X`, doi:10.1016/0169-8095(88)90035-X.

---

## Author Comment (AC2)

**Response to comments from editors and reviewers**

Part 2: Joint multifractal analysis of available wind power and rain intensity from an operational wind farm (npg-2024-6)

preprint in Nonlinear Processes in Geophysics from 02 Feb 2024
Editor's decision received on
* * *
The authors would like to thank the reviewer for evaluating the paper and providing a detailed feedback. Please find below our point-by-point response to the comments (Reviewer comments are shown in black and author responses are in blue).

**Anonymous Referee #2, 19 Mar 2024**

This study entilted « Joint multifractal analysis of available wind power and rain intensity», investigates the quantification of the effect of rainfall on wind power through the scale invariant framework of Universel Multifractals.

This manuscript is structured as follows : after an introduction part, in the part 2 the authors describe the framework of UM and JMF. In the part 3 the results of analyses of UM and JMF are presented for respectively individual and jointly data fields. The part 4 concerns a discussion part on the influence of rain type as well as that of wind direction on power production. Section 5 concludes the study and summarizes the results.

Thank you very much for taking the time to read and review our manuscript. We greatly appreciate the feedback. Please find our response to the points raised below.

*In this study, the authors propose a new parameter JMF, from UM framework, to quantify the effect of rain on wind power output. This is represented a novelty for the scientific community and can be interest the eolian energy scientific community. However, the power output analyzed are values of available power output, instead of actual power output due to the presence of biais as indicated by the authors. For the understanding, this would relevant to insert the reference explaining this point or add in the manuscript the corresponding simulations.*

Thank you. This point is addressed in Part 1 of the paper where the presence of rated value in actual threshold caused biases in estimation. To avoid this, and also due to this limitation, actual power had to be used. Though this is mentioned in section 2.3.3, with mention to part 1, we agree that a proper reference to previous paper is needed. For now, only a reference to preprint is added. Reference to final paper will be included.

*Minor Revisions*

I suggest to authors to zoom the following result figures n°3, 4, 11-14, A1 and A2.

We have increased the size of Fig. 3, 11 and A1. For the rest the size is limited by the width of the page. We believe them to appear bigger in final format since the pdf of first draft used more side margin space in general and has line numbers.

In fig. 3, K(q) curve represented in dotted red line is not visible.

Since there is no second scaling regime in the considerations, the line is redundant.

Typos : line page 16 line 358 (see section ??)..

Thank you, this is corrected now.

**References**